# The Antileukemic Effect of Xestoquinone, A Marine-Derived Polycyclic Quinone-Type Metabolite, Is Mediated through ROS-Induced Inhibition of HSP-90

**DOI:** 10.3390/molecules26227037

**Published:** 2021-11-21

**Authors:** Kuan-Chih Wang, Mei-Chin Lu, Kai-Cheng Hsu, Mohamed El-Shazly, Shou-Ping Shih, Ssu-Ting Lien, Fu-Wen Kuo, Shyh-Chyun Yang, Chun-Lin Chen, Yu-Chen S. H. Yang

**Affiliations:** 1School of Pharmacy, College of Pharmacy, Kaohsiung Medical University, Kaohsiung 807, Taiwan; dx9901mk5@gmail.com; 2Graduate Institute of Marine Biology, National Dong Hwa University, Pingtung 944, Taiwan; jinx6609@nmmba.gov.tw (M.-C.L.); fuwen@nmmba.gov.tw (F.-W.K.); 3National Museum of Marine Biology & Aquarium, Pingtung 944, Taiwan; 4Graduate Institute of Cancer Biology and Drug Discovery, College of Medical Science and Technology, Taipei Medical University, Taipei 110, Taiwan; piki@tmu.edu.tw (K.-C.H.); stly1996@tmu.edu.tw (S.-T.L.); 5Ph.D. Program for Cancer Molecular Biology and Drug Discovery, College of Medical Science and Technology, Taipei Medical University, Taipei 110, Taiwan; 6TMU Research Center of Drug Discovery, Taipei Medical University, Taipei 110, Taiwan; 7Department of Pharmacognosy, Faculty of Pharmacy, Ain-Shams University, Organization of African Unity Street, Cairo 11566, Egypt; mohamed.elshazly@pharma.asu.edu.eg; 8Department of Pharmaceutical Biology, Faculty of Pharmacy and Biotechnology, German University in Cairo, Cairo 11835, Egypt; 9Doctoral Degree Program in Marine Biotechnology, National Sun Yat-Sen University (NSYSU), Kaohsiung 804, Taiwan; m6430005@hotmail.com; 10Doctoral Degree Program in Marine Biotechnology, Academia Sinica, Taipei 115, Taiwan; 11Department of Fragrance and Cosmetic Science, College of Pharmacy, Kaohsiung Medical University, Kaohsiung 807, Taiwan; 12Department of Medical Research, Kaohsiung Medical University Hospital, Kaohsiung 807, Taiwan; 13Department of Biological Sciences, National Sun Yat-Sen University, Kaohsiung 804, Taiwan; 14Department of Biotechnology, Kaohsiung Medical University, Kaohsiung 807, Taiwan; 15Graduate Institute of Natural Products, College of Pharmacy, Kaohsiung Medical University, Kaohsiung 807, Taiwan; 16Joint Biobank, Office of Human Research, Taipei Medical University, Taipei 110, Taiwan

**Keywords:** polycyclic quinone, xestoquinone, apoptosis, anticancer, topoisomerase, reactive oxygen species (ROS), Heat shock protein 90 (HSP-90)

## Abstract

Xestoquinone is a polycyclic quinone-type metabolite with a reported antitumor effect. We tested the cytotoxic activity of xestoquinone on a series of hematological cancer cell lines. The antileukemic effect of xestoquinone was evaluated in vitro and in vivo. This marine metabolite suppressed the proliferation of Molt-4, K562, and Sup-T1 cells with IC_50_ values of 2.95 ± 0.21, 6.22 ± 0.21, and 8.58 ± 0.60 µM, respectively, as demonstrated by MTT assay. In the cell-free system, it inhibited the activity of topoisomerase I (Topo I) and II (Topo II) by 50% after treatment with 0.235 and 0.094 μM, respectively. The flow cytometric analysis indicated that the cytotoxic effect of xestoquinone was mediated through the induction of multiple apoptotic pathways in Molt-4 cells. The pretreatment of Molt-4 cells with N-acetyl cysteine (NAC) diminished the disruption of the mitochondrial membrane potential (MMP) and apoptosis, as well as retaining the expression of both Topo I and II. In the nude mice xenograft model, the administration of xestoquinone (1 μg/g) significantly attenuated tumor growth by 31.2% compared with the solvent control. Molecular docking, Western blotting, and thermal shift assay verified the catalytic inhibitory activity of xestoquinone by high binding affinity to HSP-90 and Topo I/II. Our findings indicated that xestoquinone targeted leukemia cancer cells through multiple pathways, suggesting its potential application as an antileukemic drug lead.

## 1. Introduction

Heat shock proteins (HSPs) are a group of proteins that are expressed in response to elevated temperatures or other stresses [1]. HSPs exhibit diverse functions including protein folding, translocation of organelles across membranes, assembling and disassembling of proteins, signaling transduction, degradation of misfolded proteins, and ROS generation leading to apoptosis and cell cycle arrest [2,3]. HSPs participate in the resistance to chemotherapy and radiotherapy [4]. HSP-90 has been linked to cancer, viral infection, and inflammation [5,6]. It is highly expressed in acute myeloid leukemia patients [7]. HSP-90 expression is associated with the high total leucocytic count, circulating blasts in peripheral blood and blasts in bone marrow as well as bad prognosis including relapse or death in acute myeloid leukemia (AML) patients [8]. HSP-90 inhibitors can suppress *p*-glycoprotein function, which may overcome drug resistance [9]. The combination of HSP-90 inhibitors and radiation can enhance the DNA damage effect [10]. HSP-90 inhibitors attracted attention as potential treatments of cancers [11]. Although no drug targeting HSP-90 has been approved by the FDA [12], clinical trials of HSP-90 inhibitors are currently running [13]. These inhibitors are tested against different cancers including breast, prostate, lung, malignant melanoma, chronic myeloid leukemia, and multiple myeloma [14].

Topoisomerases are another group of interesting targets in the war against cancer. There are two major families of topoisomerases including topoisomerases I (Topo I) and topoisomerases II (Topo II). Both Topo I and Topo II are targets of many anticancer agents. Camptothecins are Topo I inhibitors including topotecan and irinotecan. Topo II inhibitors include anthracyclines doxorubicin, daunomycin, epipodophyllotoxins etoposide and teniposide, amsacrine, and mitoxantrone [15,16,17,18]. Most Topo II inhibitors induce apoptosis through the induction of reactive oxygen species (ROS) and the disruption of the mitochondrial pathway [19,20]. Topo II inhibitors suppress enzyme-mediated DNA ligation, causing the accumulation of double-stranded breaks leading to cell cycle arrest and apoptosis [21]. Topo inhibitors have been used as the first line of treatment against leukemia for many years [21]. They induce chromosomal breakage, resulting in a potent anti-cancer effect [22]. Combination therapy of HSP-90 and Topo inhibitors was proposed to overcome cancer resistance and reduce side effects [23]. Some HSP-90 inhibitors were shown to interact with topo IIα resulting in a synergistic anti-proliferation effect via the inhibition of both HSP-90 and Topo II [23].

Reactive oxygen species (ROS) play important roles in the regulation of normal physiological functions such as cell cycle progression, proliferation, differentiation, migration, and cell death [24]. ROS are short-lived, highly reactive species [25]. DNA damage results in the production of most ROS [26]. HSPs regulate the inflammatory cascade by inhibiting pro-inflammatory factors, leading to the production of endogenous ROS and intrinsic cell apoptosis [27]. ROS cause cell apoptosis through several signaling pathways including mitochondrial, endoplasmic stress, and death receptor pathways [24]. In the mitochondrial pathway, ROS regulate the Bcl-2 family, induce cytochrome-*c* release from the mitochondria, induce apoptosis, and disrupt mitochondrial membrane potential (MMP) [28]. In the ER pathway, ROS regulate ER through the Bcl-2 family and Ca^2+^ release from the ER through Bax-Bak channels as the hallmark of ER stress [29]. Links between ROS and death receptor also exist. ROS regulate the TNF-R1 pathway or Fas-mediated pathway that induces cell apoptosis [30,31].

Our previous study showed that halenaquinone, a polycyclic quinone-type metabolite extracted from *Petrosia* sp. sponge, exhibited diverse biological activities, especially cytotoxicity against leukemia. There are two major natural polycyclic quinone-type metabolites, halenaquinone (HQ) and xestoquinone (XQ), isolated from *Petrosia* sp. sponge [32]. Accumulating evidence indicated that HQ is a broad-spectrum tyrosine kinase inhibitor [32]. Our previous study demonstrated the antileukemia effect of HQ in vitro and in vivo animal xenograft models. HQ acted as a potent dual inhibitor of topoisomerases I and II α with IC_50_ of 2.905 and 0.0055 µg/mL, respectively, using the cell-free system assay [32]. On the other hand, the cytotoxic activity of XQ was not investigated. Here we report that the marine active compound, XQ, suppressed the growth of human leukemia Molt 4 cells’ tumor in xenograft animal model. XQ inhibited cancer cell proliferation and induced apoptosis through ROS generation. It acted as a novel catalytic inhibitor of topoisomerases I and IIα that was distinct from all known topoisomerases poisons including etoposide, teniposide, and fluoroquinolones [33,34]. We elucidated the precise mechanism of the antileukemic activity of XQ.

## 2. Results

### 2.1. Effect of Xestoquinone on Cellular Growth of Hematologic Cancer Cells

To investigate the cytotoxic effect of XQ (Figure 1a), we first assessed the effect of XQ on the proliferation of a panel of human cancer cell lines including leukemia (Molt-4 and K562 cells) and lymphoma (Sut-T1 and U937 cells) cells. XQ exhibited cytotoxic activity against Molt-4, K562, U937, and Sup-T1 cells with IC_50_ values of 2.95 ± 0.21, 6.22 ± 0.21, 11.12 ± 0.19, and 8.58 ± 0.60 µM, respectively (Table 1). We analyzed the effect of XQ on the cell viability of leukemia and lymphoma cell lines. The results of the MTT assay showed that XQ inhibited the proliferation of Molt-4, K562, Sup-T1, and U937 cells in a dose-dependent manner. Low doses of XQ (1.96, 3.92, and 7.84 µM) significantly inhibited the proliferation of Molt-4 cells. At the highest dose (15.68 μM), XQ almost eliminated cell viability in both leukemia Molt-4 and K562 cells, resulting in 7.4 ± 2.8 and 5.6 ± 1.7% of survival rates, respectively. A similar effect was observed in both lymphoma Sup-T1 and U937 cells, resulting in 24.6 ± 2.2 and 17.8 ± 2.1% viable cells at the highest dose of XQ (15.68 µM). In contrast, XQ did not significantly suppress the cell viability of normal rat lung macrophage NR8383 using at the highest dose (15.68 µM) after 24 h (Figure 1b). These data showed that XQ is a potent antiproliferative marine metabolite against leukemia cells.

### 2.2. Xestoquinone Interacted with Multiple Targets including Topo I, Topo II, and HSP-90, as Demonstrated by Molecular Docking Analysis

Molecular docking analysis was exploited to determine the interaction between XQ 249 and its targets, Topo II and HSP-90 (Figure 2). XQ bound to the Topo II, Topo I, and HSP-90 active sites with docking scores of −26.9, −24.0, and −15.5, respectively. XQ interaction with its target can be divided into four sections based on the areas of interactions. In Topo II, Section 1 formed hydrogen bonds with the residues Asn150, Ser148, and Ser149 (Figure 2a). This was facilitated by the cyclic oxygen in the ring structure and the carbonyl oxygen that acted as a hydrogen acceptor (Figure 2b). The other three sections formed hydrophobic interactions due to the cyclic structure of XQ. Section 2 extended into the Topo II binding site and created hydrophobic interactions with various residues, such as Gly161, Gly164, and Ala167 (Figure 2b). Section 3 and Section 4 consisted of a naphthoquinone ring, which formed hydrophobic interactions with the residues Ile125 and Ser149, respectively (Figure 2b). No hydrogen bond was observed with the carbonyl oxygens in these sections. Section 3 and Section 4 were located towards the outer rim of the Topo II binding site (Figure 2a). XQ bound to the active site of DNA cleavage in Topo I and formed hydrophobic interactions (Figure 2c). Topo I residue Asn352 formed hydrophobic interactions with Section 2. Other hydrophobic interactions with DNA nucleotides sandwiched Section 3 (Figure 2d). A hydrogen bond was formed between the carbonyl in Section 3 and the residue Arg364. Interactions with Section 1 and Section 4 were not observed. Nevertheless, the interactions at Section 2 and Section 3 suggested that xestoquinone can bind to the Topo I binding site. XQ showed HSP-90 inhibition. Molecular docking analysis showed that Section 3 and Section 4 of XQ extended into the binding pocket (Figure 2e). Section 3 formed hydrophobic interactions with the residues Asn51 and Thr184. The residues Ser52 and Ala55 formed hydrophobic interactions in Section 4. The HSP-90 residue Asn51 formed part of a deep binding site pocket 3. One of the carbonyl groups in Section 3 formed a hydrogen bond with the residue Asn51 (Figure 2e). Section 2 was in the periphery of the HSP-90 binding site. This section created hydrophobic interactions with residue Lys58 (Figure 2e). Finally, a hydrogen bond was created between the Lys58 side chain and the cyclic oxygen at Section 1. Together, these interactions showed that XQ formed interactions with the Topo II and HSP-90 binding sites.

### 2.3. Xestoquinone Inhibited Topoisomerase I and II Activities

Our previous study indicated that HQ could act as a dual catalytic inhibitor and inhibited Topo I and II activities with IC_50_ of 1.19 and 0.0055 μg/mL, respectively [32]. To confirm whether XQ affected the activity of Topo I and II, we used different doses of XQ in the cell-free DNA cleavage assay. Initially, the effect of XQ on Topo I activity was studied. A cell-free DNA cleavage assay using an enzyme-mediated negatively supercoiled pHOT1 plasmid DNA was used to study this effect. As shown in Figure 3a, XQ at lower doses (0.245–1.96 μM) induced DNA relaxation in the presence of Topo I (lanes 1–4), but, at higher doses (3.92–7.84 μM), it partially inhibited the ability of Topo I to convert the supercoiled DNA to the relaxed form (lanes 5–6) with IC_50_ value of 0.24 μM. Both XQ and camptothecin equally increased the amount of the relaxed DNA; even the amount of supercoiled DNA produced by 7.84 μM of XQ was more than that produced by 10 mM of camptothecin. The potential activity of XQ on topo I would be better compared with camptothecin. We next assessed the impact of XQ on the activity of Topo II with a cell-free DNA cleavage assay. As shown in Figure 3b, XQ at lower doses (0.013–0.06 μM) induced DNA relaxation in the presence of Topo II (lanes 4–5), but, at higher doses (0.245–0.98 μM), it completely inhibited the effect of Topo II to convert the supercoiled DNA to the relaxed form (lanes 6–7) with IC_50_ value of 0.094 μM. A linear DNA strand was observed on the supercoiled pHOT1 plasmid DNA treated with etoposide (10 mM), a standard Topo II poison (lane 1). The same results were obtained using a marine secondary metabolite, halenaquinone [32]. In parallel, we examined the effect of XQ on the activity of Topo and DNA damage-repaired proteins in leukemia Molt-4 cells using Western blotting assay. The exposure to XQ (7.84 μM) induced a significant increase of phosphorylated checkpoint kinase 1 (p-Chk1), phosphorylated checkpoint kinase 2 (p-Chk2), and γ-H_2_AX expression. There was no statistical significance of XQ treatment on the expression of Topo I in a time-independent manner. Interestingly, the abolishment of Topo II expression was observed when XQ at 7.84 μM was used after 18 h (Figure 3c). These findings suggested that this marine polycyclic quinone-type metabolite, XQ, exhibited potent catalytic inhibitory activity of topo II, resulting in DNA damage.

### 2.4. Xestoquinone Inhibited Histone Deacetylases (HDACs) in Human Leukemia Molt-4 Cells

Recent reports indicated that HDACs modify topoisomerases’ activity [35]. HDACs’ expressions were determined by Western blotting assay using the Molt-4 cells’ model. A significant decrease in the expressions of HDACs including HDAC 3, HDAC 4, HDAC 6, HDAC 7, and HDAC 8 were observed with XQ (7.84 µM) treatment for 24 h. In addition, the XQ only inhibited the expression of HDAC8, and other HDACs remained unchanged in K562 cells (Figure 4a). The pan-HDACs’ activity was determined by ELISA. HDACs’ activity was blocked by XQ at different concentrations (Figure 4b).

### 2.5. Xestoquinone Induced Cell Death in Human Leukemia Cells via Death Receptor Pathway and Apoptosis

Xestoquinone treatment promoted the expression of several apoptotic markers including Annexin V externalization, PARP, caspases-3 and -7 in Molt4, K562, SupT1, and U937 leukemia cell lines (Figure 5a,b). XQ induced Fas and TRADD expressions after 1 h (Figure 5c). Moreover, the results showed that caspase-8 was activated after 6 h of drug treatment while XIAP expression was suppressed. The drug treatment activated caspases-3, -7, and -9 and PARP cleavage after 9 h (Figure 5d). These results indicated that XQ induced cell apoptosis via the activation of the death receptor pathway.

### 2.6. Xestoquinone Induced Mitochondrial Dysfunction, ER Stress, and Calcium Release in Human Leukemia Cells

The disruption of mitochondrial membrane potential (MMP) was determined by flow cytometry. MMP was significantly increased with XQ treatment in human leukemia Molt-4 and K562 cells. Compared with the solvent control, the percentage of MMP disruption in Molt-4 cells increased from 6.2% to 30.4%, 49.9%, and 88.7, respectively. In K562 cells the percentage of MMP destruction increased from 7.59% to 25.3%, 36.6%, and 63.2%, respectively (Figure 6a). The calcium release from ER was also detected. The results showed that after 60 min at XQ, the calcium ion concentration in Molt-4 and K562 cytoplasm increased significantly by 1.48 and 1.26 folds and continued for 300 min (Figure 6b). XQ affected the Bcl-2 family that controls mitochondrial and ER membrane permeabilization, and its effect was detected by Western blotting. XQ treatment significantly increased Bax and Bak levels and decreased Bcl-XL but did not affect Bcl-2 and Bim expression (Figure 6c). Our results indicated that XQ treatment increased the intracellular calcium release and accumulation (Figure 6b). To study the effect of XQ on endoplasmic reticulum stress, three major transmembrane transduction proteins, PERK, ATF 6, and IRE 1α, and ER chaperones were determined by Western blotting. The expressions of the proteins GRP94, PDI, and CHOP were also detected. The results showed that 18 h of XQ treatment inhibited the expression of PERK and IRE 1α, while the performance of Grp94 was maintained about 2 times after 6 h of XQ treatment, and the expression of CHOP increased after 9 h of XQ treatment compared to control (Figure 6d). The results suggested that the apoptotic effect of XQ involved mitochondrial dysfunction.

### 2.7. Xestoquinone Induced Topoisomerase Cleavage, Resulting in Cell Apoptosis via ROS Generation

The carboxy derivative of fluorescein, carboxy-H2DCFDA dye, was used to monitor the generation of ROS. Flow cytometry assay indicated that XQ treatment (7.84 μM) increased ROS production in Molt-4 and K562 cells by 3.26 and 1.46 folds, respectively (Figure 7a,b). To clarify the role of ROS in XQ-induced cell apoptosis, the ROS inhibitor N-acetyl cysteine (NAC) was used. NAC was pre-incubated for 1 h and XQ was added for 30 min. As shown in Figure 7b, the result of NAC treatment was similar to that of the negative control group, and the accumulation of intracellular ROS was recovered to about 1 fold. The pretreatment with NAC reduced the apoptotic cell population in Molt-4 and K562 cells from 72.1% and 62.0% to 13.7% and 11.4%, respectively. It also reduced PARP cleavage and caspase-3 activation that were induced by XQ (Figure 7c,d). NAC pretreatment reduced the cell populations affected by MMP from 96.9% and 66.3% to 15.4% and 13.6%, respectively. The increase in calcium release decreased from 1.48 and 1.26 to 0.82 and 0.99 folds (Figure 7e,f). These results indicated that NAC inhibited XQ-induced ROS production, mitochondrial dysfunction, calcium release, and apoptosis (Figure 7b–f). Topo I and II α cleavage induced by XQ was suppressed by NAC (Figure 7g).

### 2.8. Effect of Xestoquinone on Tumor Growth in Xenograft Animal Model

To assess the in vivo anti-tumor activity of XQ, the tumor growth of a human leukemia Molt-4 xenograft was determined by a xenograft mouse model. Molt-4 (1 × 10^5^) cells were inoculated subcutaneously at the right flank of female immunodeficient athymic mice. After 1 month of treatment, the tumor growth of Molt-4 cells was significantly suppressed with the administration of XQ (1 μg/g) interpretationally. The average tumor size on day 50 in the control group was 1113.9 ± 436.5 mm^3^, whereas the average tumor size in the XQ-treated group was 766.1 ± 318.7 mm^3^ (Figure 8a). The tumor size was significantly lower in the XQ-treated group as compared with the control group (*p* = 0.049), with no significant difference in the mice body weights (data not shown). At the end of the treatment, the tumor tissue was isolated and weighed. The average weights of the tumor were less in the XQ-treated group (0.40 ± 0.15 g) compared with the control group (0.72 ± 0.30 g) (Figure 8b). The tumor volume and weight were significantly reduced in the XQ groups by 43.86% and 31.22% as compared with the control group. No remarkable differences were observed in the mice’s body weights, as well as no histopathological changes of the heart, kidney, liver, lung, and spleen being detected by H & E staining (Figure 8c). Additionally, the HDAC expression in the xenograft tumors was also confirmed. The results showed that HDAC-1, -3, and -8 were significantly suppressed. These results demonstrated the anti-tumorigenic effect of the XQ in the in vivo xenograft model.

### 2.9. Interaction of Xestoquinone and HSP-90 Protein

Protein thermal shift assay was used to identify the effect of compounds on the stability and affinity of the specific targeted protein [36]. Fluorescence-based protein thermal shift assay was applied to verify whether XQ interacted with HSP-90 using a real-time PCR system, which incrementally heated samples over the temperature gradient (from 25 °C to 95 °C) and simultaneously determined fluorescence intensity [37]. As shown in Table 2 and Figure 9, there were significant differences between the melting temperature (Tm) values of HSP-90 protein (47.33 ± 2.53 °C) and HSP-90 protein treated with 17-AAG (54.88 ± 3.44 °C and 55.76 ± 3.46 °C) at 106.71 and 426.85 μM, respectively (Figure 9a). On the other hand, the Tm values of HSP-90 protein treated with XQ (84.25 ± 1.90 °C and 84.95 ± 0.86 °C) at 196.32 and 785.28 μM, respectively, shifted to a higher temperature compared with the control, with Tm value of 47.33 ± 2.53 °C (HSP-90 protein only) (Figure 9b). The results indicated that XQ interacted with HSP-90 with a higher affinity than 17-AAG. 

## 3. Discussion

Xestoquinone (XQ) was isolated from *Petrosia* sp. sponge through bio-guided fractionation and exhibited interesting biological activities, as indicated in a previous report [32]. In this study, XQ demonstrated cytotoxic activity against leukemia cancer cell lines. XQ inhibited cell proliferation of leukemia cell lines in vitro (Figure 1) and in vivo (Figure 8). XQ treatment-induced topoisomerases I and II cleavage accompanied DNA damage (Figure 3). HDACs’ activity is known to be regulated by topoisomerases [35]. HDACs’ protein levels were decreased with XQ treatment in different cells lines, and the HDACs’ activity was also significantly suppressed (Figure 4). Moreover, the HDACs’ proteins inhibitory activity of XQ was also confirmed in the in vivo xenograft model (Figure 8e).

XQ induced anti-proliferation in leukemia cancer cell lines through apoptosis via multiple mechanisms. These mechanisms included the death-receptor pathway, in which XQ promoted FADD, Fas, and TRADD protein expression (Figure 5). XQ also affected the mitochondria pathway by disrupting MMP and Bcl-2 family (Figure 6a,c). It influenced the ER stress pathway by inducing calcium release and RE stress markers including the suppression of IRE1a, PERK, and ATF6 expression and the promotion of Grp94, PDI, and CHOP expression (Figure 6b,d). Moreover, XQ induced ROS generation (Figure 7a,b). The hallmarks of apoptosis including annexin V translocated to the outer leaflet on the plasma, promotion of RARP cleavage, caspase-3 activity, death receptor expression, MMP, and calcium release were suppressed using the ROS inhibitor, NAC (Figure 7c–f). Interestingly, the cleavage of Topo I and Topo IIα were also blocked by NAC (Figure 7g). Those results suggested that XQ induced cell apoptosis via ROS generation. The XQ-induced ROS generation may come from different sources including DNA damage, release from mitochondria, or the inhibition of heat shock proteins (HSPs). The XQ-induced DNA damage and MMP change were blocked by NAC (Figure 7). These results indicated that the XQ-induced ROS generation does not come from DNA damage or disruption of MMP but possibly from the inhibition of HSPs.

Our molecular docking analysis revealed that xestoquinone can bind to the Topo II, Topo I, and HSP-90 binding sites. This suggested that XQ has the potential of acting as a multi-targeting inhibitor. XQ structure can be divided into four sections based on its interactions with the target protein binding sites (Figure 2). Section 1 showed a preference for hydrogen bond interaction. This was observed with hydrogen bond formation with Topo II (residues Ser148, Ser149, and Asn150) and HSP-90 (residue Lys58) (Figure 2). In addition, protein thermal shift assay was used to offer a high-throughput method to identify the interactions between protein targets and low-molecular-weight compounds in an incremental temperature using specific fluorescent dyes [38,39,40]. The fluorescence of HSP-90 with XQ or 17-AAG was significantly inhibited in a dose-dependent manner. With the semi-automated analysis of customized software, the Tm value of HSP-90 with XQ (84.25 and 84.95 °C) was shifted to higher Tm than 17-AAG (54.88 and 55.76 °C), suggesting that XQ might bind at more than one site and not bind only to the folding stats (Figure 9a,b). Section 2 and Section 4 formed hydrophobic interactions with the binding sites of Topo II and HSP-90. The cyclic structure of XQ facilitated the hydrophobic interactions. Section 3 was sandwiched by hydrophobic interactions in all three targets. Of the three targets, XQ formed fewer interactions with Topo I (Figure 2c–d). The lack of a hydrogen bond at Section 1 and hydrophobic interactions at Section 4 may explain its weaker inhibitory activity toward Topo I. Nevertheless, the presence of a hydrogen bond at Section 3 and hydrophobic interactions that sandwiched the structure suggested that XQ can bind to Topo I. The molecular analysis showed that XQ can interact with Topo II, Topo I, and HSP-90, leading to their inhibition with a cell-free system and protein thermal shift assay, respectively. Moreover, the HSP-90 inhibition induced ROS generation, which promoted cell apoptosis via multiple apoptotic pathways (Figure 10).

## 4. Materials and Methods

### 4.1. Chemicals and Biological Materials

Cell lines were obtained from the American Type Culture Collection (ATCC, Manassas, VA, USA). The cells were maintained at 37 °C in 5% CO_2_ (humidified atmosphere). A growing medium of RPMI 1640 medium was used, which was supplemented with glutamine (2 mM), antibiotics (100 μg/mL of streptomycin and 100 units/mL of penicillin), and fetal calf (10%) serum. Antibodies against cleaved caspases-3 and -7 and cleaved-PARP, caspase-7, p-chk1 (Ser345), p-chk2 (Thr68), HDAC1, HDAC3, HDAC4, HDAC6, HDAC7, Bak, Bim, IRE1α, PDI, FADD, Fas, and TRADD were obtained from Cell Signaling Technologies (Beverly, MA, USA). Streptomycin, fetal calf serum (FCS), trypan blue, RPMI 1640 medium, and penicillin G were purchased from GibcoBRL (Gaithersburg, MD, USA). Antibodies for PARP, caspase-3, Bcl-2, Bax, Bcl-XL, PERK, ATF6, CHOP, Chk1, Chk2, γ-H2AX, HDAC 8, topoisomerase I, and topoisomerase IIα were purchased from Santa Cruz Biotechnology (Santa Cruz, CA, USA). GRP94 antibody was purchased from Enzo life sciences (Taipei, Taiwan). TNFRSF1A polyclonal antibody was purchased from ABclonal Technology (Taipei, Taiwan). Anti-mouse and rabbit IgG peroxidase-conjugated secondary antibody was purchased from Pierce (Rockford, IL, USA). Annexin V-FITC/PI (propidium iodide) stain was obtained from Strong Biotech Corporation (Taipei, Taiwan). The 3-(4,5-Dimethylthiazol-2-yl)-2,5-diphenyl-tetrazolium bromide (MTT) and dimethyl sulfoxide (DMSO) and all other chemicals were obtained from Sigma-Aldrich (St. Louis, MO, USA). Carboxy derivative of fluorescein (carboxy-H_2_DCFDA), Fluo-3, and JC-1 cationic dye were obtained from Molecular Probes and Invitrogen technologies (Carlsbad, CA, USA). ECL Western blotting detection kits and Hybond ECL transfer membrane were obtained from Amersham Life Sciences (Amersham, UK).

### 4.2. Preparation of Xestoquinone Stock Solution

Xestoquinone was separated from the marine sponge *Petrosia* sp. The spectroscopic data (1D and 2D NMR) of the isolated compound were compared to the previous report, and the chemical structure of XQ was confirmed [41]. The compound was dissolved in dimethyl sulfoxide (DMSO) at a concentration of 30 μM, and a series of dilutions were prepared before use.

### 4.3. Assay of Pan-Histone Deacetylase Activity

According to the manufacturer’s protocol (BPS Biosciences, San Diego, CA, USA), the compound (including the carboxylic and the hydroxamic acids) was screened for its pan-HDAC inhibitory activity using an HDAC inhibitor drug screening kit at different concentrations of the compound for 30 min at 37 °C. The deacetylation of the substrate sensitized the substrate; so, the next step, the treatment with 10 μL of lysine developer, produced a chromophore to stop the reaction for 30 min at 37 °C. The fluorescence was recorded at excitation = 350–380 nm and emission = 440–460 nm, using a spectrophotometer (Biotek synergy, Winooski, VT, USA).

### 4.4. Annexin V/PI Apoptotic Assay

The externalization of phosphatidylserine (PS) and membrane integrity were quantified using an annexin V-FITC staining kit [42]. In brief, 10^6^ cells were grown in 35-mm diameter plates and were labeled with annexin V-FITC (10 μg/mL) and PI (20 μg/mL) before harvesting. After labeling, all plates were washed with a binding buffer and harvested. Cells were resuspended in the binding buffer at a concentration of 2 × 10^5^ cells/mL before analysis by flow cytometer FACS-Calibur (Becton-Dickinson, San Jose, CA, USA) and CellQuest software. Approximately 10,000 cells were counted for each determination.

### 4.5. Determination of ROS Generation, Calcium Accumulation, and MMP Disruption

These assays were performed as described previously [43]. MMP disruption, calcium accumulation, and ROS generation were detected with JC-1 cationic dye (5 μg/mL), the fluorescent calcium indicator (Fluo 3, 5 mM), and the carboxy derivative of fluorescein (carboxy-H_2_DCFDA, 1.0 mM), respectively. In brief, the treated cells were labeled with a specific fluorescent dye for 30 min. After labeling, cells were washed with PBS and resuspended in PBS at a concentration of 1 × 10^6^ cells/mL before analysis via flow cytometry.

### 4.6. Western Blotting Analysis

Cell lysates were prepared by treating the cells for 30 min in RIPA lysis buffer, 1% Nonidet P-40, 0.5% sodium deoxycholate, 0.1% sodium dodecyl sulphate (SDS), 1 mM sodium orthovanadate, 100 μg/mL phenylmethylsulfonyl fluoride, and 30 μg/mL aprotinin) (all chemicals were from Sigma) [32,43]. The lysates were centrifuged at 20,000× *g* for 30 min, and the protein concentration in the supernatant was determined using a BCA protein assay kit (Pierce). Equal amounts of proteins were, respectively, separated by 7.5%, 10%, or 12% of SDS-polyacrylamide gel electrophoresis and then were electrotransferred to a PVDF membrane. The membrane was blocked with a solution containing 5% non-fat dry milk TBST buffer (20 mM Tris-HCl, pH 7.4, 150 mM NaCl, and 0.1% Tween 20) for 1 h and washed with TBST buffer. The protein expressions were monitored by immunoblotting using specific antibodies. These proteins were detected by an enhanced chemiluminescence kit (Pierce, Rockford, IL, USA).

### 4.7. Immunofluorescence Analysis

After treatment with the tested compound, cells were fixed with 4% paraformaldehyde in 50 mM HEPES buffer (pH 7.3) for 30 min and permeabilized for 20 min with 0.2% Triton X-100 in PBS (pH 7.4). To prevent non-specific protein binding, cells were incubated with 5% BSA in PBS containing 0.05% Triton X-100 (T-PBS) for 1 h at room temperature. Cells were then incubated with the primary antibodies (1:500) for 2 h and with the secondary antibodies (Alexa Fluor 586-conjugated goat anti-mouse IgG (H + L) (Life Technologies, Carlsbad, CA, USA) diluted at 1:1000 for 1 h at room temperature. After washing with PBS, cells were observed under an FV1000 confocal laser scanning microscope (Olympus, Tokyo, Japan).

### 4.8. Human Leukemia Molt-4 Cells’ Xenograft Animal Model

The establishment of nude mice with xenografts was performed as described previously [32,43]. Six-week-old male immunodeficient athymic mice were purchased from the National Laboratory Animal and Research Center (Taipei, Taiwan). All animals were maintained under standard laboratory conditions (temperature 24–26 °C, 12–12 h dark-light cycle) and fed with a laboratory diet and water. This study was approved by the Animal Care and Treatment Committee of Kaohsiung Medical University (IACUC Permit Number 105139). All experiments were conducted in strict accordance with the recommendations in the Guide for the Care and Use of Laboratory Animals of the National Institutes of Health, and all efforts were made to minimize animal stress/distress. Molt-4 cells (1 × 10^6^) resuspended in 0.2 mL PBS were injected s.c. into the right flank of each mouse, and tumor growth was monitored every day. Fourteen days after tumor cell injection, mice with confirmed tumor growth were randomly divided into two groups. XQ (1.8 μg/g) was intraperitoneally administered to the treatment group, and the control group received solvent only. XQ was administrated every-other day for 66 days. Animals were sacrificed by carbon dioxide. Tumor size was measured three times a week using calipers, and the tumor volumes were calculated according to the standard formula: width^2^ × length/2.

### 4.9. Histopathological Evaluation

Histopathological examination was performed by Prof. Jiunn-Wang Liao of the Animal Disease Diagnostic Center (ADDC), College of Veterinary Medicine, National Chung Hsing University, to examine and evaluate the histopathologic results report. The severity was graded according to the method described by Shackelford et al. [44]. In short, the tumor, heart, kidney, liver, lung, and spleen tissues of nude mice in the control group (n = 7) and the XQ group (n = 7) were divided into three equal parts (proximal, middle, and distal) with formalin-fixed tissue, embedded in paraffin, sectioned (2 µm), and hematoxylin and eosin (H&E) were used.

### 4.10. Molecular Docking Assay

The molecular docking software, LeadIT [45], was used to perform the docking analysis. The HSP-90 (PDB ID: 3VHC), Topo II (PDB ID: 1ZXM), and Topo I (PDB ID: 1K4T) crystal structures were obtained from the Protein Data Bank [46]. LeadIT was used to prepare the protein for docking by removing water molecules and the co-crystal ligand. A radius of 10 Å from the co-crystallized ligand was defined as the binding site. An enthalpy and entropy docking approach was used. The maximum solutions for each iteration and each fragmentation were set to 300. Default settings were used for all other parameters.

### 4.11. Protein Thermal Shift Assay

The manufacturer’s protocol (Thermo Fisher Scientific Baltics, Vilnius, Lithuania) recommended HSP-90 to be used as the protein (1 μg/μL/reaction) and different doses of XQ (1 μL) to be used as the ligand. Then, 5 μL of thermal shift buffer, 2.5 μL of 8× Sypro Orange fluorescent dye, and 10.5 μL of water (the final volumes of 20 μL) were added to each well using a Multidrop Combi Reagent Dispenser (Thermo Fisher Scientific, Waltham, MA, USA). The plate was then sealed with MicroAmp Optical Adhesive Film (Applied Biosystems, Foster City, CA, USA) and spun down. The reaction was mixed at the bottom of the plate. The plate was heated from 25 to 99 °C with the heating rate of 1 °C/min and was measured with the ROX dye channel (Ex/Em: 490 nm/530 nm) on the Step One Plus Real-Time PCR instrument (Applied Biosystems, Foster City, CA, USA). XQ and 17-AAG with different concentrations were determined in triplicates. The melting temperatures (Tm) and thermal profiles were conducted using Protein Thermal Shift Software (version 1.3, Applied Biosystems, Foster City, CA, USA).

### 4.12. Statistics

The results were expressed as mean ± standard deviation (SD). Comparison in each experiment was performed using an unpaired Student’s *t*-test. A *p*-value of less than 0.05 was considered to be statistically significant.

## 5. Conclusions

The treatment of a series of hematological cancer cells with the polycyclic quinone-type metabolite, xestoquinone, induced mitochondrial dysfunction, ROS generation, and ER stresses resulting in apoptosis. In the cell-free system assay, XQ showed effective inhibition of topoisomerases I and II activity with IC_50_ values of 0.235 and 0.094 μM, respectively. Molecular docking and Western analysis blotting indicated that the catalytic inhibitory activity of XQ was attributed to its high binding affinity to Topo II, Topo I, and HSP-90 binding sites. The fluorescence-based protein thermal shift assay was performed to identify the interaction of XQ and HSP-90. This suggested that XQ has the potential as a multi-target inhibitor. Combination therapies or multi-target drugs could maximize the therapeutic potential of anticancer drugs [47,48]. The pretreatment of Molt-4 cells with N-acetylcysteine (NAC) reduced XQ-induced disturbance of mitochondrial membrane potential (MMP) and apoptosis and retained the expression of Topo I and II. Compared with the solvent control in the nude mouse xenograft model, the administration of XQ (1 μg/g) significantly reduced tumor growth by 31.2%. Our results showed that XQ can interact with Topo I, Topo II, and HSP-90, leading to their inhibition. It induced ROS production by inhibiting the expression of HSP-90. XQ promoted cell apoptosis through a variety of apoptotic pathways, suggesting its potential application as an antileukemic drug lead.

## Figures and Tables

**Figure 1 molecules-26-07037-f001:**
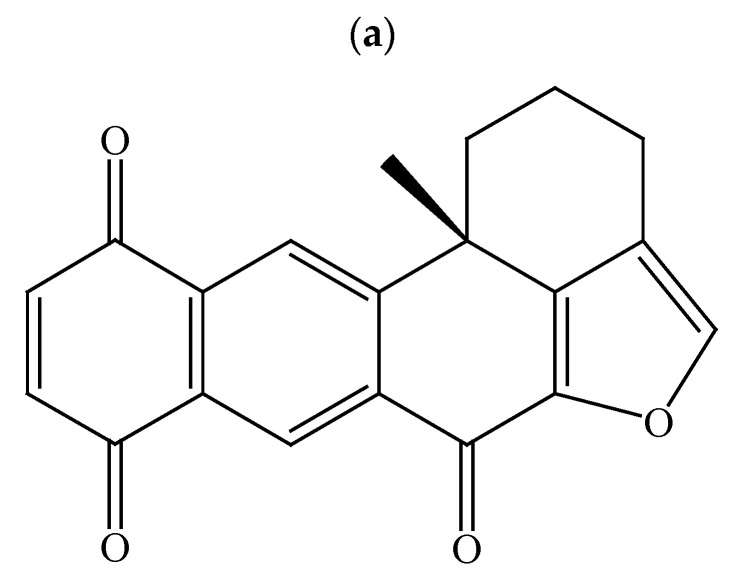
Effect of xestoquinone (XQ) on cellular viability in the in vitro cellular model. (**a**) The structure of the marine polycyclic quinone metabolite XQ isolated from the ethyl acetate (EtOAc) extract of *Petrosia* sp. sponge. (**b**) Human leukemia (Molt-4 and K562) cells and lymphoma (Sup-T1 and U937) cells as well as normal rat NR8383 macrophages were treated with XQ at different doses for 24 h. The viability was determined by the MTT assay. The results are presented as means ± SD of three independent experiments; * *p* < 0.05, ** *p* < 0.01, and *** *p* < 0.001.

**Figure 2 molecules-26-07037-f002:**
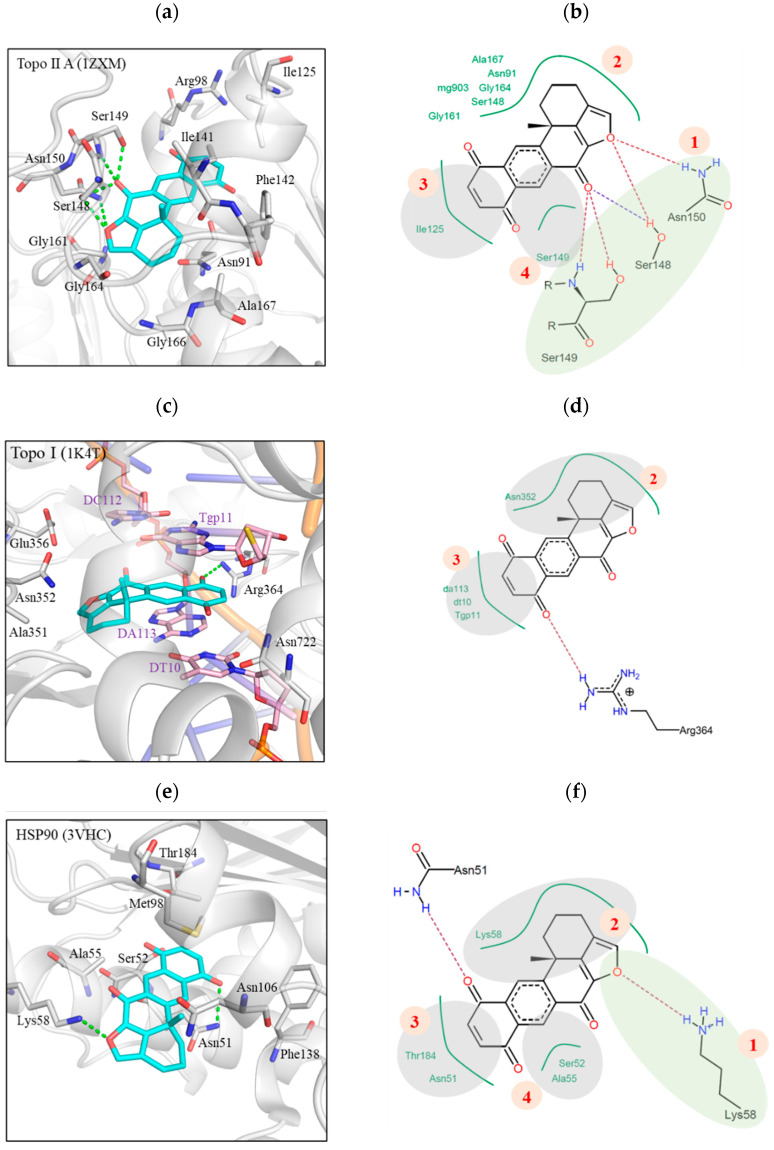
Docking results of XQ with Topo II, Topo I, and HSP-90 binding sites (**a**) XQ (blue) docked in Topo II (gray). (**b**) The 2D interaction map of xestoquinone in Topo II binding site. (**c**) XQ (blue) docked in Topo I (gray). (**d**) The 2D interaction map of XQ docked in Topo I. (**e**) XQ (blue) docked in the HSP-90 (gray) binding site. (**f**) A 2D interaction map between XQ and HSP-90. Hydrogen bonds in 3D figures are denoted by green lines. The 2D interaction map was generated by LeadIT. Common hydrophobic (gray) or hydrogen bonding (green) areas are highlighted. Hydrogen bonds are denoted by dashed lines. Hydrophobic interactions are 333, represented as green lines. Residues are labeled as shown.

**Figure 3 molecules-26-07037-f003:**
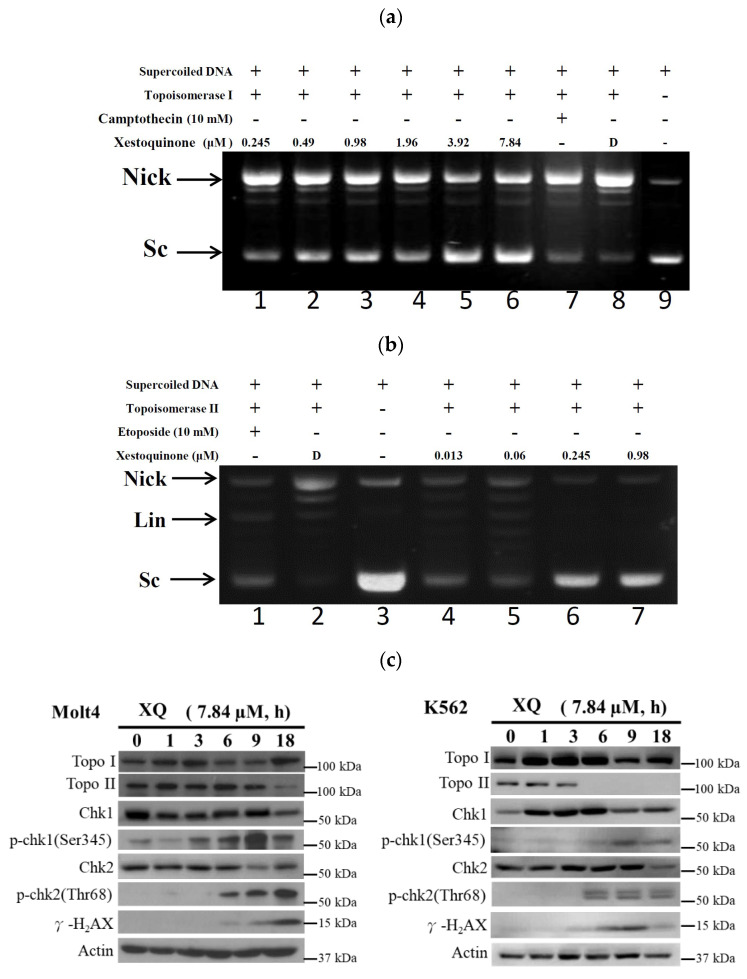
Effect of XQ on Topo I and II activity. (**a**) The effect of XQ on Topo I mediated supercoiled pHOT1 plasmid DNA relaxation with the cell-free system. Lanes 1–6: XQ (0.245, 0.49, 0.98, 1.96, 3.92, and 7.84 μM); Lane 7: Positive control, camptothecin (10 mM); Lane 8: Plasmid DNA + Topo I + solvent control (induction of DNA relaxation); Lane 9: Negative control plasmid DNA (supercoiled DNA). Nick, nicked DNA; Sc, supercoiled DNA. (**b**) The effect of XQ on Topo II mediated supe-coiled pHOT1 plasmid DNA relaxation with the cell-free system. Lane 1: Positive control, etoposide (10 mM), as Topo II poison (induction of linear DNA); Lane 2: Plasmid DNA + topo IIα + solvent control (induction of DNA relaxation); Lane 3: Negative control plasmid DNA (supercoiled DNA). Lanes 4–7: XQ (0.013, 0.06, 0.245, and 0.98 μM). Nick, nicked DNA; Sc, supercoiled DNA; Lin, linear DNA. (**c**) Effect of XQ on the expression of topo IIα and DNA damage-related proteins of Molt-4 cells and K562 cells. Molt-4 and K562 cells were treated with XQ (7.84 μM) for the indicated time intervals. The protein expression was analyzed with Western blotting. Actin was used as the loading control.

**Figure 4 molecules-26-07037-f004:**
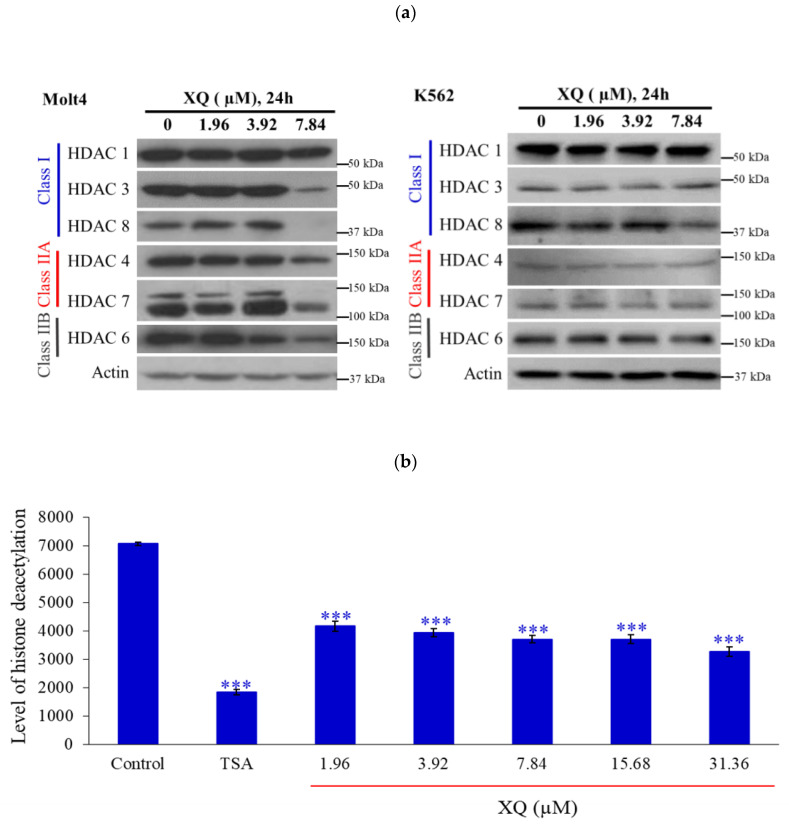
Xestoquinone inhibited HDACs’ expressions in human leukemia Molt-4 and K562 cells. Molt-4 and K562 cells were treated with XQ for 24 h and the whole-cell lysates were collected. (**a**) HDACs’ protein expressions were detected by Western blotting. (**b**) In the presence or absence of different doses of HDAC inhibitor XQ and TSA (positive control), a specific HDAC fluorescent substrate composed of acetylated lysine side chains is incubated with pan-HDAC enzyme before adding lysine Acid developer to detect pan-HDACs’ activity by ELISA. The results are presented as means ± SD of three independent experiments; *** *p* < 0.001. Actin was used as the loading control.

**Figure 5 molecules-26-07037-f005:**
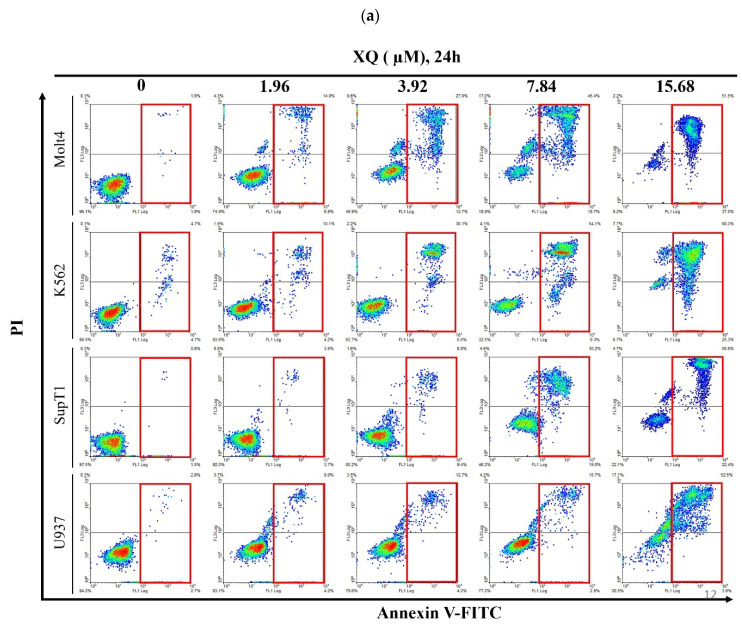
Xestoquinone induced cell apoptosis in Molt4, K562, SupT1, and U937 human leukemia cells. Leukemia cells were treated with XQ for 24 h. (**a**) The annexin V externalization was detected by flow cytometry. (**b**) Caspases-3 and -7 and PARP protein levels were detected by Western blotting. (**c**) The protein levels of death receptors, including FADD, Fas, and TRADD, were detected by Western blotting. (**d**) The effect of XQ on the expression of apoptosis-related proteins was determined by Western blotting analysis. Actin was used as the loading control.

**Figure 6 molecules-26-07037-f006:**
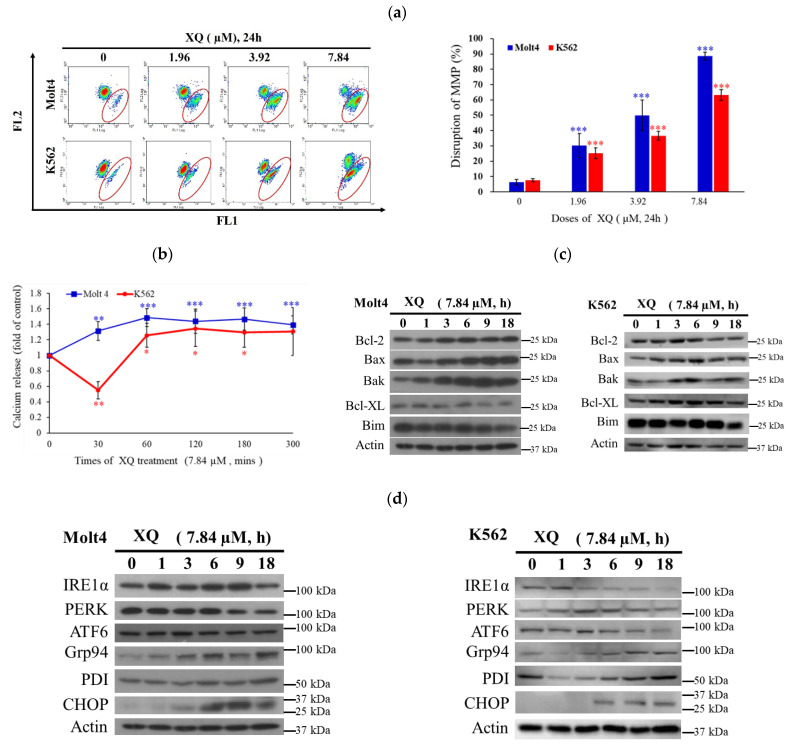
The apoptotic effect of XQ on leukemia cells is mediated through mitochondrial dysfunction and ER stress. Leukemia cells were treated with XQ for 24 h. (**a**) XQ induced changes in MMP that were detected by flow cytometry. (**b**) XQ induced calcium release from ER into the cytosol. (**c**) The leukemia cells were treated with XQ treatment for 24 h and whole-cell lysates were collected. XQ regulated the Bcl-2 family, and the effect was evaluated by Western blotting. (**d**) Leukemia cells were treated with XQ for 24 h and whole-cell lysates were collected. The markers of ER stress were detected by Western blotting. Actin was used as the loading control. The results are presented as means ± SD of three independent experiments; * *p* < 0.05, ** *p* < 0.01, and *** *p* < 0.001.

**Figure 7 molecules-26-07037-f007:**
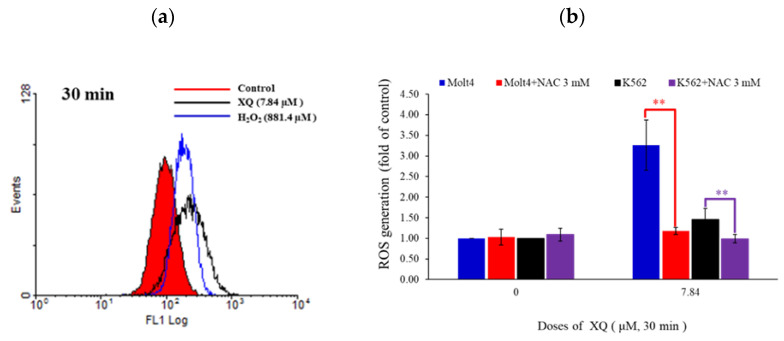
XQ-induced apoptosis in leukemia cells involved the promotion of ROS generation. (**a**) Leukemia Molt-4 cells were treated with XQ for 30 min. The concentration of ROS was detected by flow cytometry. H_2_O_2_ was used as the positive control. (**b**) Leukemia Molt-4 and k562 cells were pretreated with NAC for 30 min followed by XQ treatment for 30 min. The concentration of ROS was detected by flow cytometry. Molt-4 leukemia cells were pretreated with NAC for 30 min followed by XQ treatment for 24 h. (**c**) The annexin-V expression was detected by flow cytometry. (d) Whole-cell lysates were collected. The effect of XQ on apoptosis-related proteins was evaluated by Western blotting. (**e**) The leukemia cells were pretreated with NAC for 30 min followed by XQ treatment for 24 h. The changes in MMP were detected by flow cytometry. (**f**) The cells were treated with NAC for 30 min followed by XQ treatment for 1 h, and the calcium release from ER into the cytosol was evaluated. (**g**) Leukemia cells were pretreated with NAC for 30 min followed with XQ for 24 h, and the whole-cell lysates were collected. The protein expressions of Topo I and Topo IIα were detected by Western blotting; * *p* < 0.05, ** *p* < 0.01, and *** *p* < 0.001. Actin was used as the loading control.

**Figure 8 molecules-26-07037-f008:**
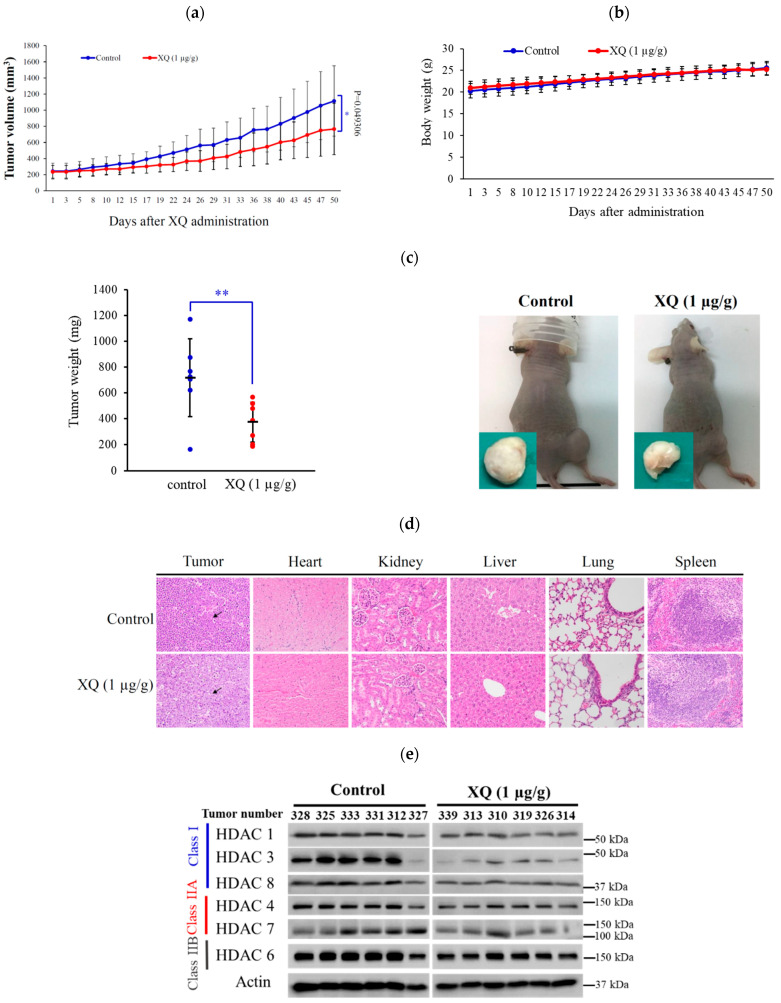
Effect of XQ on tumor growth in the in vivo tumor xenograft animal model. Tumor-bearing nude mice were intraperitoneally injected with the solvent control (DMSO, n = 7) and XQ (1 μg/g, n = 7) for 50 days. (**a**) Tumor volumes were measured every-other day, and the results were expressed as mean ± SD. * Significantly different from the control groups at * *p* < 0.05; ** *p* < 0.01. (**b**) The body weight was measured every-other day, and the results were expressed as mean ± SD. (**c**) Each blue (control group) or red (XQ-treated group) circle represents the weight of one tumor, respectively. Histogram of the tumor weight from the control group and XQ-treated group. Values are expressed as mean ± SD. ** Significantly different from the control groups at *p* = 0.001. Representative photos of the subcutaneous tumors were collected after treatment with the solvent only (left) or with XQ (right) for 50 days. (**d**) Histopathological findings of organs after the treatment with DMSO and XQ in the subcutaneous xenograft of Molt-4 cells in mice. Tumor cells expressed round cells’ shape, high nucleic/cytoplasm ratio with high mitosis (arrow) in the DMSO and XQ-treated groups; no significant lesions in the heart, kidney, liver, lung, and spleen were detected in the XQ group stained with H&E stain (400×). (**e**) The HDACs’ proteins’ expressions in tumor xenograft were detected by Western blotting. Actin was used as the loading control.

**Figure 9 molecules-26-07037-f009:**
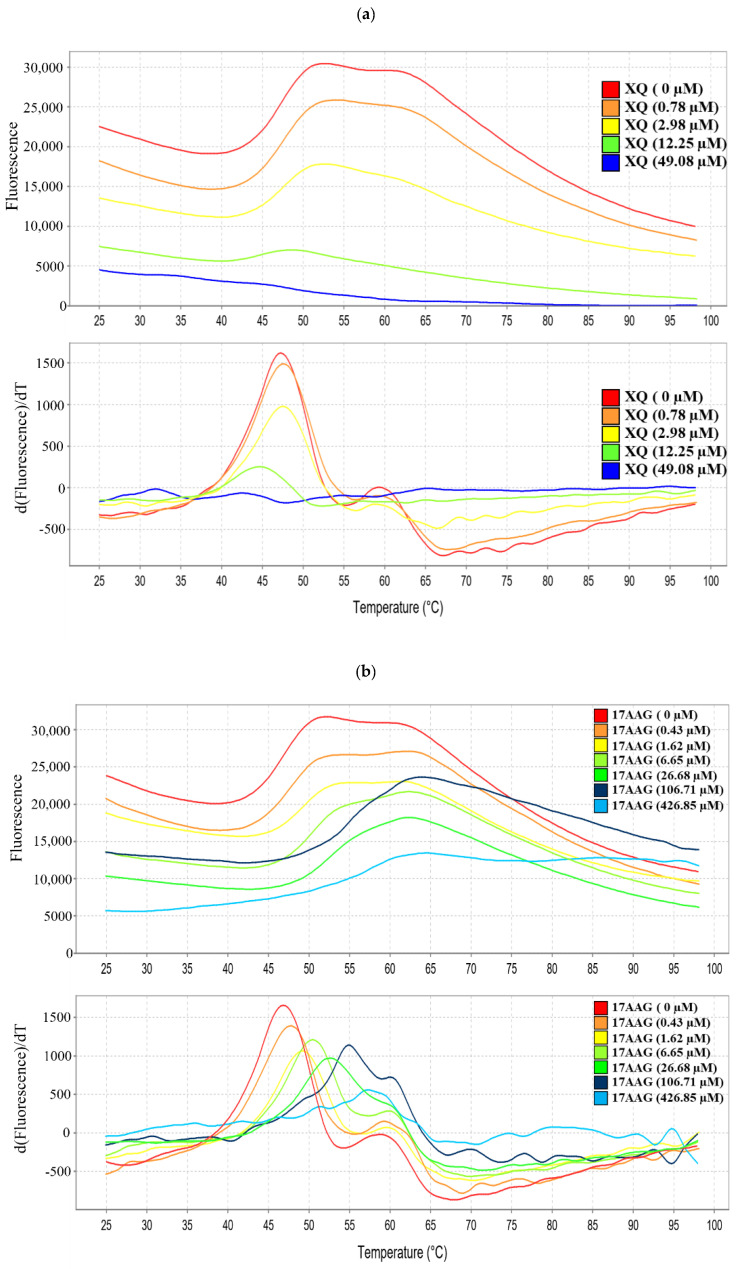
Melting profile of HSP-90. Melting profile of HSP-90 (1 μg) with different doses of (**a**) xestoquinone (XQ) and (**b**) 17-AAG. The upper figures show the thermal denaturation profiles of HSP-90 proteins with XQ or 17-AAG, respectively. The lower figures show the profiles of the deriv-ative of fluorescence emission as a function of temperature (dF/dT) in HSP-90 treated with XQ or 17-AAG, respectively.

**Figure 10 molecules-26-07037-f010:**
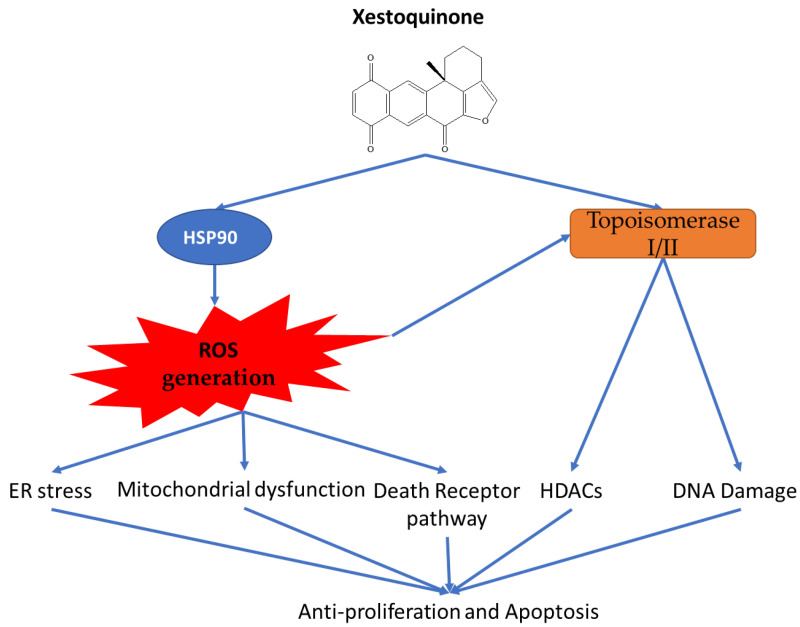
Xestoquinone induced apoptosis in leukemia cells via multiple mechanisms.

**Table 1 molecules-26-07037-t001:** Cytotoxic effect of XQ against cancer cell lines and non-cancer cell lines.

Cancer Types	Cell Lines	XQ IC_50_ (µM)
Leukemia cancer cells	Molt-4	2.95 ± 0.21
	K562	6.22 ± 0.21
Lymphoma cancer cells	Sup-T1	8.58 ± 0.60
	U937	11.12 ± 0.19
Normal rat macrophage	NR8383	>30

**Table 2 molecules-26-07037-t002:** Melting temperature (Tm) of HSP-90 in the presence of 17-AAG and XQ.

Sample	Tm Values (°C)
HSP-90	47.33 ± 2.53
HSP-90+17-AAG (1.25 μg/mL; 106.71 μM)	54.88 ± 3.44 ***
HSP-90+17-AAG (5 μg/mL; 426.85 μM)	55.76 ± 3.46 ***
HSP-90+XQ (1.25 μg/mL; 196.32 μM)	84.25 ± 1.90 ***
HSP-90+XQ (5 μg/mL; 785.28 μM)	84.95 ± 0.86 ***

Values represent the mean ± SD for three independent experiments; *** *p* < 0.001, compared with the control.

## Data Availability

Not applicable.

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
