# Peer review of "The Antileukemic Effect of Xestoquinone, A Marine-Derived Polycyclic Quinone-Type Metabolite, Is Mediated through ROS-Induced Inhibition of HSP-90"

_molecules, 2021, doi:10.3390/molecules26227037_

Round 1
Reviewer 1 Report
The major part of my previous suggestions has be done by the authors. In this revised version in my opinion paper can be published.
Author Response
Reply to the Reviewers’ Comments
Reviewer 1
Comments and Suggestions for Authors
The major part of my previous suggestions has be done by the authors. In this revised version in my opinion paper can be published.
Response: We would like to express our deep appreciation for the reviewer's comments. We are grateful for her/his constructive remarks and for recommending to accept our paper.
Reviewer 2 Report
The re-submitted manuscript represents much improved version of the original work presented by Wang and co-authors earlier this year. Despite of addressing most of my specific comments there are two main criticisms which were ignored. Re-submission allows authors to introduce extensive modifications to the manuscript beyond the point-by-point response to reviewer’s comments (expected at revision), regrettably, Wang and co-authors decided not to take advantage of this opportunity.
First, I’m still perplexed by the disbalance between the great amount of time and effort dedicated to the experimental part of the project and an obvious lack of attention associated with the manuscript preparation. Authors need to keep in mind that reader, including reviewers and editors, will be influenced by the formal qualities of their work when judging scientific merits of the reported findings. Lack of attention to details mentioned in my previous review is still present and countless errors and mistakes throughout the manuscript need to be fixed before publication.
A handful of examples (by far not a full list):
Figure 2, panel (b): Image error.
Figure 4: Western quantification panels seem to be swapped.
Moreover, the information about western blot quantification in the figure legend is insufficient and should be described in detail in the Methods section. Including normalization method, number of independent replicates, and - importantly – what has been done to ensure that measured values reside within the linear range of the method. Since western blot is a semi-quantitative method, representative western blots without quantification panels are acceptable.
Western blot quantification panels variably do or do not have their own panel labels. If authors decide to keep them in the figures, I suggest to label all of them.
Figure 4 legend: K562 cells are not mentioned.
Figure 5 legend: “(a) The annexin V externalization was detected by flow cytometry. The whole-cell lysates were collected.” The second sentence makes no sense, probably copied-over from some other place.
Figure 5 legend: No description of panel (d).
Line 132: p-chk1 and p-chk2 denote most likely phosphorylated forms of the proteins. It wouldn’t harm to explain the acronym in the text. It is also unclear why authors chose to spell unphosphorylated protein with a capital “C” and phosphorylated in small case both in text and figures.
Second, I need to express my doubts about the native speaker’s energy invested in reading the manuscript. If authors decide to submit new version of this work for publication, I urge them strongly to read it carefully before sending the manuscript to the editor. Please find a couple examples below illustrating my point.
Line 333: “Molecular ducking assay” Despite of the line has been corrected (chapter numbering), this obvious typo was neglected.
Figure 1 legend: “The chemical structure of marine polycyclic quinone-type metabolite, XQ, was isolated from Petrosia sp. sponge.” Either correct or describe in detail how a chemical structure can be obtained from a sponge.
Line 102: “XQ exhibited potent cytotoxic against Molt-4, K562,…” Missing word.
Lines 151-152: “Moreover, the activity of caspase 8 starts at 6 hours, while the XIAP decrease, caspase 9, 7, 3 activity and PARP cleavage were appeared immediately (Figure 5d).” In addition to the poor grammar, “immediate” means without any delay, not after 9 hours.
Figure 8 legend: “Effect of XQ on tumor growth in the in vivo human leukemia Molt-4 cell tumor xenograft animal model.” Excessive wording.
And so on and so forth.
Numerous punctuation mistakes.
Overall, my verdict is rejection. Please correct the errors in the manuscript (not only the examples highlighted by reviewers as examples), read it carefully again, and re-submit. Not doing it means wasted time of reviewers and editors followed by yet another rejection.
Author Response
Reply to the Reviewers’ Comments
Reviewer 2
Comments and Suggestions for Authors
The re-submitted manuscript represents much improved version of the original work presented by Wang and co-authors earlier this year. Despite of addressing most of my specific comments there are two main criticisms which were ignored. Re-submission allows authors to introduce extensive modifications to the manuscript beyond the point-by-point response to reviewer’s comments (expected at revision), regrettably, Wang and co-authors decided not to take advantage of this opportunity.
First, I’m still perplexed by the disbalance between the great amount of time and effort dedicated to the experimental part of the project and an obvious lack of attention associated with the manuscript preparation. Authors need to keep in mind that reader, including reviewers and editors, will be influenced by the formal qualities of their work when judging scientific merits of the reported findings. Lack of attention to details mentioned in my previous review is still present and countless errors and mistakes throughout the manuscript need to be fixed before publication.
A handful of examples (by far not a full list):
Figure 2, panel (b): Image error.
Response: We are sincerely sorry for this mistake. We are thankful for highlighting this error. We corrected figure 2 in the manuscript.
Figure 4: Western quantification panels seem to be swapped.
Response: We are sorry for this error. We are thankful to the reviewer for indicating this error. We corrected the Western quantification panel in the revised manuscript.
Moreover, the information about western blot quantification in the figure legend is insufficient and should be described in detail in the Methods section. Including normalization method, number of independent replicates, and - importantly – what has been done to ensure that measured values reside within the linear range of the method. Since western blot is a semi-quantitative method, representative western blots without quantification panels are acceptable.
Response: We are thankful for the reviewer's remark. We added the description of the Western blot quantification assay in the “Materials and Methods” section.
Western blot quantification panels variably do or do not have their own panel labels. If authors decide to keep them in the figures, I suggest to label all of them.
Response: We are grateful for the reviewer's comment. We added the molecular weight of the marker to the Western blot results in the revised manuscript.
Figure 4 legend: K562 cells are not mentioned.
Response: We are grateful for the reviewer's remark. We added the K562 cell description to the figure 4 legend in the revised manuscript.
Figure 5 legend: “(a) The annexin V externalization was detected by flow cytometry. The whole-cell lysates were collected.” The second sentence makes no sense, probably copied-over from some other place.
Response: We are sincerely sorry for this mistake. Following the reviewer’s remark, we corrected the sentence in the revised manuscript.
Figure 5 legend: No description of panel (d).
Response: We are thankful for the reviewer's comment. We added the description to the figure legend 5d in the revised manuscript.
Line 132: p-chk1 and p-chk2 denote most likely phosphorylated forms of the proteins. It wouldn’t harm to explain the acronym in the text. It is also unclear why authors chose to spell unphosphorylated protein with a capital “C” and phosphorylated in small case both in text and figures.
Response: We are grateful for the reviewer’s remark. We corrected the capitalized words and explained the acronym in the text of the revised manuscript.
Second, I need to express my doubts about the native speaker’s energy invested in reading the manuscript. If authors decide to submit new version of this work for publication, I urge them strongly to read it carefully before sending the manuscript to the editor. Please find a couple examples below illustrating my point.
Line 333: “Molecular ducking assay” Despite of the line has been corrected (chapter numbering), this obvious typo was neglected.
Response: We are sincerely sorry for this mistake. We corrected the statement in the revised manuscript.
Figure 1 legend: “The chemical structure of marine polycyclic quinone-type metabolite, XQ, was isolated from Petrosia sp. sponge.” Either correct or describe in detail how a chemical structure can be obtained from a sponge.
Response: We are sincerely sorry for this mistake. Following the reviewer's suggestion, we corrected the figure legend in the revised manuscript.
Line 102: “XQ exhibited potent cytotoxic against Molt-4, K562,…” Missing word.
Response: We are sincerely sorry for this mistake. We corrected the sentence in the revised manuscript.
Lines 151-152: “Moreover, the activity of caspase 8 starts at 6 hours, while the XIAP decrease, caspase 9, 7, 3 activity and PARP cleavage were appeared immediately (Figure 5d).” In addition to the poor grammar, “immediate” means without any delay, not after 9 hours.
Response: We are sincerely sorry for this mistake. We corrected the sentence in the revised manuscript.
Figure 8 legend: “Effect of XQ on tumor growth in the in vivo human leukemia Molt-4 cell tumor xenograft animal model.” Excessive wording.
Response: We are sincerely sorry for this mistake. We rephrased the sentence in the revised manuscript.
And so on and so forth.
Numerous punctuation mistakes.
Overall, my verdict is rejection. Please correct the errors in the manuscript (not only the examples highlighted by reviewers as examples), read it carefully again, and re-submit. Not doing it means wasted time of reviewers and editors followed by yet another rejection.
Response: We are sincerely sorry for any mistake in the manuscript. We never intended to waste the editor’s nor the reviewers’ time. We are sincerely grateful for the editors’ and reviewers’ efforts and thorough reading of our manuscript and their constructive comments. The reviewers’ remarks significantly improved the quality of the manuscript. We thank all reviewers for their valuable remarks and comments.
Reviewer 3 Report
I am satisfied with the authors' response to my comments, and I believe that this paper is now suitable for the publication in the journal, Molecules.
Author Response
Reply to the Reviewers’ Comments
Reviewer 3
Comments and Suggestions for Authors
I am satisfied with the authors' response to my comments, and I believe that this paper is now suitable for the publication in the journal, Molecules.
Response: We are thankful for the reviewer's remark. We are grateful for accepting our submitted paper.
Reviewer 4 Report
The authors have addressed some of my comments. My last two comments are:
- For the immunoblot results, molecular weight markers should be shown and indicate the molecular size of the target bands. Also, uncropped full-size immunoblot membrane should be shown in the supplements.
- The authors are advised for a final check of the manuscript by native English speakers to ensure there is no typos and grammatical mistakes.
Author Response
Reply to the Reviewers’ Comments
Reviewer 4
Comments and Suggestions for Authors
The authors have addressed some of my comments. My last two comments are:
For the immunoblot results, molecular weight markers should be shown and indicate the molecular size of the target bands. Also, uncropped full-size immunoblot membrane should be shown in the supplements.
Response: We are thankful for the reviewer’s comment. We added the molecular weight of the marker to the Western blot results to the revised manuscript. Since the Western blotting membrane of three independent experiments will result in overwhelming data, the cropped Western blot results were presented in the supplementary information.
The authors are advised for a final check of the manuscript by native English speakers to ensure there is no typos and grammatical mistakes.
Response: We are thankful for the reviewer's comment. We consulted a native English speaker to recheck the whole manuscript for any typos or grammatical errors.
Round 2
Reviewer 2 Report
Overall, the latest version of the manuscript represents an improvement when compared to the previous text. However, some of my concerns were not addressed properly. First, authors chose to keep western quantification panels in the manuscript, but description of measures taken to maintain measured values within the linear range of the method is still missing.
The new line in Methods “…and the semi-quantitative analysis of bands was performed using ImageJ digital imaging processing software…” makes little sense. In general, western blot is described as a semi-quantitative method, because at number of the procedure steps linearity of the assay may be compromised especially when HRP/ECL detection is used. Therefore, in cases when quantification is needed, a serial dilution of a positive control is used to demonstrate that none of the detected bands exceeded detection limits of the method. This is a good laboratory practice approach to western blot quantification.
Second, despite of clearly stating that mistakes listed in my review are just examples indicating how little attention was paid to manuscript preparation and do not represent a complete list of flaws, nothing else than those highlighted errors was corrected. My personal motivation is to help others to improve their science and provide a feedback about presented results and their interpretation. It is very frustrating when I am presented with manuscript submitted by authors who can’t be bothered (repeatedly) to proof-read what they want to publish. Trust me, I have better things to do than list all grammar errors, missing legends, or obvious confusions. Please be so kind and make sure that your future manuscripts are in much better shape before you submit them for publication, my fellow reviewers will appreciate that.
Chk1 and Chk2 are spelled with capital “C”.
Author Response
Reply to the Reviewers’ Comments
Manuscript Number: molecules-1438735
Review 2
Comments and Suggestions for Authors
Overall, the latest version of the manuscript represents an improvement when compared to the previous text. However, some of my concerns were not addressed properly. First, authors chose to keep western quantification panels in the manuscript, but description of measures taken to maintain measured values within the linear range of the method is still missing.
The new line in Methods “…and the semi-quantitative analysis of bands was performed using ImageJ digital imaging processing software…” makes little sense. In general, western blot is described as a semi-quantitative method, because at number of the procedure steps linearity of the assay may be compromised especially when HRP/ECL detection is used. Therefore, in cases when quantification is needed, a serial dilution of a positive control is used to demonstrate that none of the detected bands exceeded detection limits of the method. This is a good laboratory practice approach to western blot quantification.
Response: We highly appreciate the reviewer’s comment. Our WB results were obtained using the traditional photography with the x-ray film, which is indeed difficult to make sure whether the measured values are in the linear range. As our results have been clearly present by figures and we did not use or compare any of the measured values in the text, we decided to remove all the quantification data to avoid any potential misinterpretation and confusion to readers.
Second, despite of clearly stating that mistakes listed in my review are just examples indicating how little attention was paid to manuscript preparation and do not represent a complete list of flaws, nothing else than those highlighted errors was corrected. My personal motivation is to help others to improve their science and provide a feedback about presented results and their interpretation. It is very frustrating when I am presented with manuscript submitted by authors who can’t be bothered (repeatedly) to proof-read what they want to publish. Trust me, I have better things to do than list all grammar errors, missing legends, or obvious confusions. Please be so kind and make sure that your future manuscripts are in much better shape before you submit them for publication, my fellow reviewers will appreciate that.
Response: We thank the reviewer for devoting his/her valuable time in reviewing our manuscript. According to the Editor’s suggestion, the manuscript will be sent to the English editing service when accepted.
Chk1 and Chk2 are spelled with capital “C”
Response: Thanks for pointing out the error. We have corrected it in the revised manuscript.
This manuscript is a resubmission of an earlier submission. The following is a list of the peer review reports and author responses from that submission.
Round 1
Reviewer 1 Report
In this paper authors report biological evaluation of xestoquinone on a series of hematological cancer cell lines. The antileukemic effect of xestoquinone was evaluated in vitro and in vivo. Authors evidenced an interesting multitaget activity of this compound, because it showed antiproliferative activity on leukemic cells (without effect on healt cells), apoptotic effect, ROS generation and action of Topo I and II. Also in vivo xenograft model was performed, as well as molecular docking simulations.
Therefore paper is complete regarding biological evaluation; only minor revisions are necessary (see file in attachement).
Additional revisions:
- Authors could insert a table with reported IC50 values for antiproliferative activity regarding Figure 1;
- Please specify if XQ used for these test has been bought or isolated or others…;
- Please add an explanation regarding the used concentrations in all test performed (why they used these concentrations and not 0.1, 0.5 micromolar, 1, 10, 20 micromolar as usual?);
- , line 63 ref 13. please inserted references regarding clinical trials;
- Please add references regarding multitarget compounds as anticancer/antinflammatory agents in the conclusions.

Reviewer 2 Report
In the manuscript titled “The Antileukemic Effect of Xestoquinone, A Marine-Derived Polycyclic Quinone-Type Metabolite, is Mediated through ROS Induced Inhibition of HSP-90” Wang and co-authors present their findings on Xestoquinone (XQ) describing molecular interactions of the compound with target molecules, as well as in vitro and in vivo effects of the drug. Although it is an interesting piece of work, the manuscript suffers from a number of shortcomings leading me to suggest rejection in the current version. However, authors should consider finishing some missing experiments, re-analyzing existing datasets, and re-writing the manuscript. A new submission seems to be the right path forward here.
First and foremost, I urge authors to consider re-organizing results section. Current order of the reported experiments can be much improved. To present the story in a more reader-friendly format, I suggest opening with the structure and molecular docking analysis, moving towards in vitro effects, and concluding on the high note with in vivo data.
Second, the language used is very difficult to comprehend and either native speaker or a colleague highly proficient in English should be invited to assist with the final version of the document.
Third, attention to detail needs to be improved. From a number of obvious typos to both missing and excessive sections and references (e.g. info about a HIF antibody in materials and methods sections, but no HIF detection is to be located throughout the manuscript, immunofluorescence section in the materials and methods, etc.).
Fourth, overinterpretation of presented results. For example, pg. 8, line 188: “These results indicated that XQ induced cell apoptosis via the death receptor pathway. “ More experiments are needed to support such a strong statement.
Below please find my other comments and concerns. These are not ordered by importance. Authors are encouraged to use them as a guidance how to improve the manuscript rather than a complete list of flaws.
Please consider unification of the font size and style in your figures. Placement of panel labels is unusual and should be changed.
Figure 1B: It is unclear how statistics was calculated. Plotting all lines separately may help. Why control has no error bars? It is unlikely that NR8383 cells treated with high doses of XQ display significantly decreased viability compared to control. At least this is what the text says on pg. 3, lines 123-124.
Pg. 5, line 153. Chk1 and Chk2 do not cause DNA damage but are activated in response to DNA damage.
Concentrations of Camptothecin and Etoposide (10mM) should be confirmed (typo?).
Figure 3B: Box plot, preferably with the beeswarm/scatter option will provide much better service to present important piece of data.
Histology – not mentioned in materials and methods, how many samples were analyzed, how it was scored.
Were xenograft samples analyzed for HDACs downregulation? If this is a XQ hallmark effect, it should be documented in the tumor as well.
There is plenty of space in the figure for documenting no change in animal body weight (mentioned in the text).
ELISA is missing in the methods section.
Page 8, line 182. The statement about Annexin V is incorrect. However, it is explained correctly in the methods. Similarly, Figure 5 legend.
XQ seems to be either fluorescent or it increases autofluorescence of cells. (Figure 5A) How this was compensated for in experiments where fluorescence is measured?
Figure 5A: K562 plots need to be re-analyzed.
Figure 6A: Gate used to calculate fraction of cells with MMP collapse seems to be moving among panels. This is impermissible manipulation and data must be re-analyzed.
Figure 6B: Both positive and negative controls are missing.
Figure 7A XQ/autofluorescence may be an issue here. Please show fluorescence of both populations with no H2DCFDA staining.
Molecular docking: Authors should consider providing some quantitative information from the docking analysis and compare the values to other compounds with known inhibitory effect which bind to the same pocket.
Reviewer 3 Report
In this manuscript, the authors demonstrated the antileukemic effects of a major natural polycyclic quinone-type metabolite, xestoquinone (XQ) isolated from Petrosia sp., and show the possible mechanism on the antileukemic effects of XQ. The authors demonstrate that XQ inhibits topoisomerases I (Topo I) and topoisomerases II (Topo II) activities in vitro, and reactive oxygen species (ROS) production is required for XQ induced apoptosis in leukemia cells. Although inhibition activity of xestoquinone against Topo I has been already reported by other researchers [1], the other data shown in this study seem to be novel, and I believe that this paper is interesting to the reader of the journal, Molecules.
- Bae MA, Tsuji T, Kondo K, Hirase T, Ishibashi M, Shigemori H, Kobayashi J. Inhibition of Mammalian Topoisomerase I by Xestoquinone and Halenaquinone. Biosci Biotechnol Biochem. 1993 Jan;57(2):330-1. doi: 10.1271/bbb.57.330. PMID: 27314795.
However, involvement of HSP-90 in XQ-induced apoptosis is not well demonstrated in this manuscript. The docking simulation data do not demonstrate that XQ actually binds to Topo I, Topo II and heat shock protein-90 (HSP-90) under physiological conditions, only predict its possibility. There are no data for binding activity of XQ and HSP-90 in this manuscript, such as Kd value. Therefore, it is not clarified that XQ actually inhibits HSP-90 activity. This is my major concern about this paper.
Other specific points are listed below.
Comment #1: The binding energy obtained as a score should be provided in the molecular docking simulation results.
Comment #2: Involvement of HSP-90 on antileukemic effects of XQ is suggestive. The authors should describe about this more clearly in this manuscript. In addition, involvement of HSP-90 should not be included in the title of manuscript.
Comment #3: The authors claimed that xestoquinone induced cell death is caused through the death receptor mediated apoptosis signaling pathway, because xestoquinone treatment induces Fas and TRADD expressions. However, the authors did not investigate the caspase specifically activated by death receptor, caspase-8 activation in this study. Therefore, the data shown in this study does not demonstrate that the death receptor mediated signaling pathway is actually activated after treatment with xestoquinone. The authors should modified explanation in this manuscript.
Comment #4: The method to obtain XQ used in this study was not described in the Materials and Methods section.
Comment #5: Figure 5 and section 2.4, the information for the cell line used in this section should be provided.
Comment #6: The abbreviations, “Topo I" and "Topo II" are not defined in the Abstract section. In addition, the definition of the abbreviation, "MMP" is not required in the Abstract section. The words, "mitochondrial membrane potential" are only appeared once in the Abstract section.
Reviewer 4 Report
In the manuscript entitled " The antileukemic effect of Xestoquinone, a marine-derived polycyclic quinone-type metabolite, is mediated through ROS induced inhibition of HSP-90”, investigators attempted to assess the effects and study mechanism of Xestoquinon in treating leukemia cell lines. It is an interesting study, however, the in-vitro and in-vivo studies were not done with expected scientific rigor. In this regard, the manuscript cannot be accepted in current format.
Specific comments:
In Fig.1 b, results for the cytotoxic effects of Xestoquinone (XQ) on the proliferation of three prostate cancer cell lines (LN-cap, PC-2 and D145) were missing in the graph, but mentioned in the text (Line 116-117). Authors should present it in the results.
In Fig.2a and 2b, the loading of DNAs in all the lane should be measured and equally loaded. Especially in Fig.2b, it seems the negative control lane (lane 3) had higher amount of DNA loaded. In lane 5, besides nicked, linear and supercoild DNA band, there are multiple other bands can be seen. Investigators should perform this experiment again.
In Fig.2c, authors should include total Chk1 and Chk2 protein as control. Moreover, for the p-chk1 and p-chk2 protein, authors should indicate what is the specific phosphorylation site are recognized by these antibodies, such as Phospho-Chk1 (Ser280). Similar, in Fig. 5a and 5b, 7d, authors should include the full length PARP, caspase-3, caspase-7.
Since investigators attempted to investigate the effect and mechanism of XQ in leukemia, all the in-vitro experiments should be performed consistently using both Molt-4 and K562 cell lines (Fig.2c, Fig. 4, Fig. 6d, etc).
For Fig.5, investigators should indicate which type of human leukemia cells they used for the experiment. Is it the leukemia cells purified from patient or is it leukemia cell line (Molt-4)? It is not mentioned in the text of results section as well as the section of material and methods.
In Fig. 3, the volume of tumors between DMSO control and XQ group are not significant different until day 50 (p<0.049306) due to the overlapping of the error bar within these two groups. At the same time, there are significant difference in tumor weight (p<0.01). Authors should discuss the differences between these two results. Since measure tumor volume might be hard in the early stage in the xenograft model, it might be easier to quantify the tumor volume by using luciferase stable expressing Molt-4 cells to generate the xenograft mice model and monitor the tumor growth by using IVIS system.
The relation between XQ and HSP-90 is weak. Authors only perform bioinformatic based molecular docking analysis. In order to validate XQ binding to HSP-90, investigator should perform biophysics experiments (fluorescence based thermal shift assay, spectrofluorimetry) to examine their hypothesis.